# A NEURAL METHOD FOR GOAL-ORIENTED DIALOG SYSTEMS TO INTERACT WITH NAMED ENTITIES

## ABSTRACT

Many goal-oriented dialog tasks, especially ones in which the dialog system has to interact with external knowledge sources such as databases, have to handle a large number of Named Entities (NEs). There are at least two challenges in handling NEs using neural methods in such settings: individual NEs may occur only rarely making it hard to learn good representations of them, and many of the Out Of Vocabulary words that occur during test time may be NEs. Thus, the need to interact well with these NEs has emerged as a serious challenge to building neural methods for goal-oriented dialog tasks. In this paper, we propose a new neural method for this problem, and present empirical evaluations on a structured Question Answering task and three related goal-oriented dialog tasks that show that our proposed method can be effective in interacting with NEs in these settings.

## 1 INTRODUCTION AND PROBLEM DESCRIPTION

*Goal-oriented* dialog systems (Wen et al., 2016; Bordes & Weston, 2016a; Dhingra et al., 2017) are those in which the system tries to achieve an explicit goal by the end of the dialog. Examples include restaurant reservation, flight ticket booking, and course selection advising. One common aspect of many of these goal-oriented dialog systems is their need to interact with a large number of Named Entities (NEs). This is especially true in cases where the system has to interact with external knowledge sources such as DataBases (DB), as is the case in all of the examples mentioned above. NEs in these systems include restaurant names, place names, phone numbers, addresses, time, faculty names, course numbers, etc.

Recently, there has been a lot of interest in building neural methods for goal-oriented dialog systems. The presence of and the need to interact with large numbers of Named Entities in these tasks has emerged as a serious challenge to achieving good performance. There are different ways in which past work has tried to handle NEs in these neural systems. One straightforward way is to add each and every NE (including those in the DB) to the vocabulary, which has been evaluated for only synthetic or small tasks (Neelakantan et al., 2015). For real world tasks with large DBs, this causes an explosion in the vocabulary size and hence the number of parameters to learn. There is also the problem of not being able to learn *good* representations for individual NEs, as individual NEs (e.g. a particular phone number) generally occur only a few times in a dataset.

Another approach that has been proposed in the literature is to encode all the NEs with random representations and keep them fixed throughout (Yin et al., 2015), but here we lose the meaning associated with the neural embeddings and risk their representations interfering and correlating with those of others in unexpected ways. There is another simple way in which NEs are handled in many real world systems, which is to first recognize the NEs with either NE taggers (Finkel et al., 2005) or entity linkers (Cucerzan, 2007; Guo et al., 2013; Yang & Chang, 2015), and then replace them with NE-type tags. For example, all location names could be replaced with the tag *NE_location* and all the course numbers could be replaced with the tag *NE_course_number*. This prevents the explosion in vocabulary size, but the system loses the ability to distinguish and reference different NEs of the same type. If there are several location names mentioned in a dialog, the system loses its ability to distinguish and use them separately as needed in future utterances in the dialog. In addition to this, there is also the possibility of new NEs arising during the test time. In fact many of the Out Of Vocabulary (OOV) words that arise during test time in many Natural Language Processing (NLP) tasks are NEs.

In this paper, we propose a simple idea for neural methods to interact with NEs that handles all the aforementioned issues, including robustness to OOV NEs during test time. The core idea is to not include any of the NEs in the vocabulary, but rather to create a neural embedding for them on the fly when the agent actually encounters them and then use these representations to retrieve and use the actual NE value whenever required. We demonstrate our idea first on a simple structured Question Answering (QA) task and then on three related goal-oriented dialog tasks that are extended versions of some of the dialog bAbI tasks proposed in Bordes & Weston (2016b). We use a multiple-attention based neural retrieval mechanism to retrieve items from a DB table. The results suggest that our proposed way of handling NEs can be very useful in these tasks compared to not handling them separately.

## 2 DETAILS OF PROPOSED SOLUTION

Consider a neural dialog system participating in a dialog with a user. It has a predefined vocabulary obtained from the training data by excluding all NEs. When a user makes an utterance, the sentence encoder (e.g. Recurrent Neural Network (RNN)) of the dialog system processes that utterance, word by word. A Named Entity Recognizer (NER) is used to classify if a given word in the user utterance is a NE or not (possibly along with its NE type). In most goal-oriented dialog systems that interact with DB, these NEs come from the DB. So, there might not be a need for a separate NER, as the NEs, along with type information, can be obtained easily by referring to the DB. The knowledge of which words are NEs and their types can be very useful for goal oriented dialog tasks. For words that are part of the vocabulary, their neural embeddings can be obtained from the encoding matrix to give to the sentence encoder. If the word is a NE, then it will not be part of the vocabulary and hence will not have a neural embedding in the encoding matrix. In such cases, the dialog system uses its knowledge of the dialog so far, the current utterance so far and the NE-type (e.g. *NE_course_number*) of the encountered NE to generate a neural embedding for it. This generated embedding is used for the NE by the sentence encoder while encoding the sentence. It is also stored in a separate table called the *NE-Table*. A new empty *NE-Table* is used for each individual dialog. The *NE-Table* is populated with key-value pairs, where the key is the embedding generated by the dialog system and the value is the actual NE (e.g. *EECS 545*) encountered.

Here is an example instantiation of the idea where the sentence encoder is a simple Recurrent Neural Network (RNN). Figure 1 shows the process associated with the equations below as an instantiation of the idea in a Hierarchical Recurrent Encoder Decoder (HRED) model from Serban et al. (2016). Let $(X_t, Y_{t-1})$ be the user and system utterance at time step $t$ and $t-1$ respectively. Let $(x_{t,1}, x_{t,2}, ..., x_{t,i}, ..., x_{t,N})$ be the $N$ words in the user utterance $X_t$. The following are the equations associated with encoding a single word $x_{t,i}$ of the user sentence.

$$
\begin{aligned}
\text{if} \quad & \text{Is\_NE}(x_{t,i}) == True : \\
& \hat{x}_{t,i} = \text{NE\_type}(x_{t,i}) \\
& \hat{z}_{t,i} = W_{zx}^{enc} \hat{x}_{t,i} \\
& z_{t,i} = z_{t,i}^{ne} = W_{zd} h_{t-1}^d + W_{zh} h_{t,i-1}^{x,enc} + W_{zz} \hat{z}_{t,i} \\
\text{else} : & \\
& z_{t,i} = W_{zx}^{enc} x_{t,i} \\
h_{t,i}^{x,enc} &= \sigma(W_{hh}^{enc} h_{t,i-1}^{x,enc} + W_{hz} z_{t,i} + b_h)
\end{aligned}
\tag{1}
$$

where, Is_NE($x_{t,i}$) outputs $True$ if $x_{t,i}$ is a NE and NE_type($x_{t,i}$) gives the NE type of $x_{t,i}$ (e.g. NE_type(EECS 545) = *NE_course_number*. Note that, though NEs are not part of the vocabulary, their NE-type tags are and hence will have a embedding in the encoding matrix $W_{zx}^{enc}$. For words that are NEs, the representation of the dialog so far ($h_{t-1}^d$), the sentence representation of the current user utterance so far ($h_{t,i-1}^{x,enc}$) and the NE-type ($\hat{x}_{t,i}$) are used to generate a neural embedding ($z_{t,i}^{ne}$) on the fly and is stored in the *NE-Table* as key along with the NE $x_{t,i}$ associated with it stored as the value.

When the dialog system wants to refer/get back to this NE value in the future, it can do so by generating a key to match the keys in the *NE-Table* and then retrieve the corresponding value (e.g. *EECS 545*) and use it. For example, it can refer to a NE that it came across earlier in the dialog from the *NE-Table*, and use that in its system utterance (output sentence) or also to match (exact) over

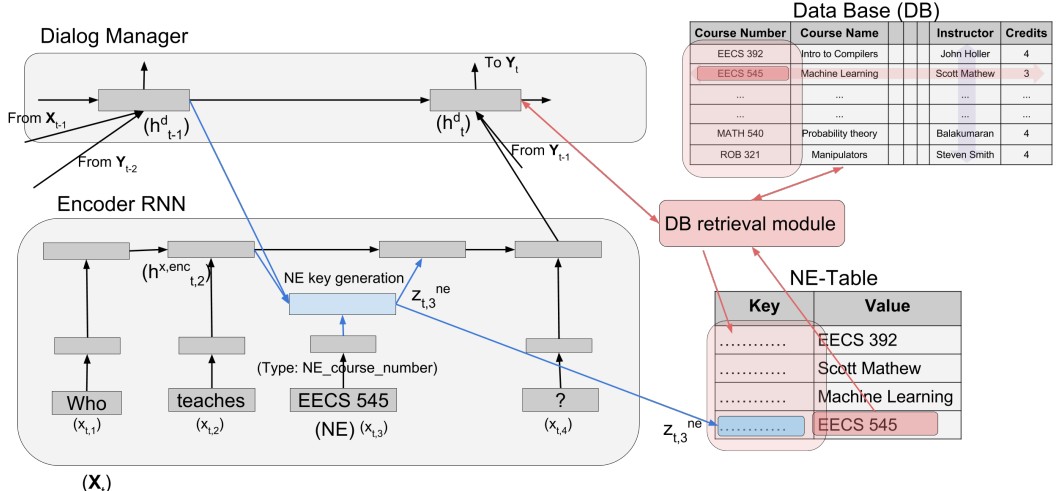

Figure 1: Instantiation of the idea in a HRED (Decoder is not shown in the figure). When the encoder RNN encounters a NE, the representation of the dialog so far ($h^d_{t-1}$), the sentence representation of the current user utterance so far ($h^{x,enc}_{t,i-1}$) and the NE type information are used to generate a neural embedding ($z^{ne}_i$) on the fly and is stored in the *NE-Table* as key along with the NE $x_{t,i}$ associated with it stored as the value. Here the DB retrieval system generates a key to match the keys in the *NE-Table* to retrieve the value (e.g. *EECS 545*) it is interested in. It can for example match (exact) over an attribute's (e.g. *Course Number*) values in an external DB (along with other required actions) to retrieve information from the DB (e.g. the name of the faculty who teaches the course with course number *EECS 545*).

an attribute's (e.g. *Course Number*) values in an external DB (along with other required actions) to retrieve information from the DB (e.g. the name of the faculty who teaches the course with course number *EECS 545*) (the specific action performed with the NE retrieved depends on the choice of the natural language generator and the DB retrieval mechanism).

The system utterance, just like the user utterance, is again processed word by word. In cases of presence of a NE in the system utterance (e.g. name of a faculty), a new representation is generated on the fly and stored in the *NE-Table* along with its value. Thus, all and only the NEs that have appeared in that particular dialog so far will be present in the *NE-Table* associated with that dialog, with their generated neural embedding keys and actual NE values.

The Dialog system learns to generate representations for the NEs as they come in, such that the representations have relevant and enough information (depending on the downstream task from which gradient signals come) which would allow it to match and retrieve them when required later in the dialog (e.g. for its system utterance or for its interaction with DB and so on).

## 3 EXPERIMENTS AND RESULTS

We evaluate our idea on two types of tasks - a simple structured Question-Answering (QA) task and goal-oriented dialog tasks, which are extended versions of some of the dialog bAbI tasks from (Bordes & Weston, 2016b). All these tasks involve interaction with DB. For all our experiments, we use a simple multiple-attention based neural retrieval mechanism which can use the NE-table idea. There are also other neural mechanisms proposed recently such as Yin et al. (2015) and Dhingra et al. (2017) which could be modified and used along with our idea for interaction with the DB.

### 3.1 MULTIPLE-ATTENTION BASED NEURAL RETRIEVAL MECHANISM

In this paper, we perform tasks on datasets where the information is present in a single database table, where each row corresponds to a new entity of interest and the columns of the table correspond to

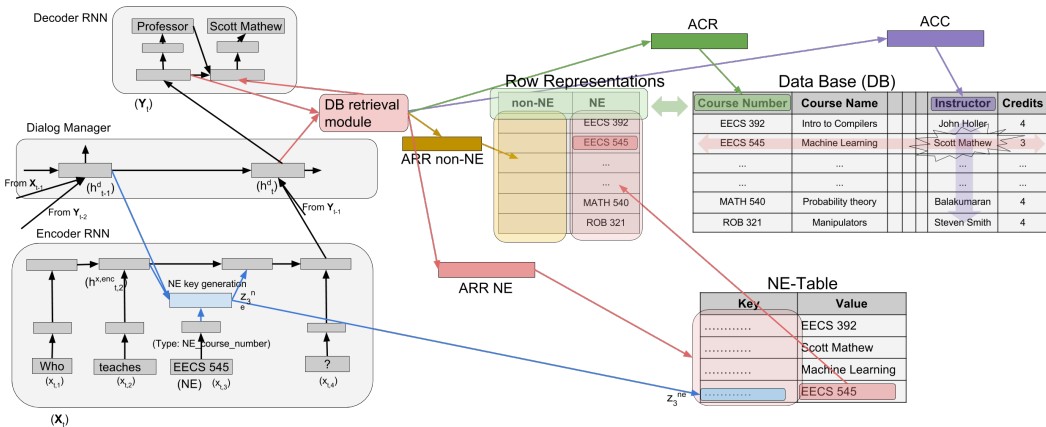

Figure 2: Multiple-attention based neural retrieval mechanism. When the encoder RNN encounters a NE, it generates a key representation for it and stores it in the *NE-Table*. When the dialog manager/decoder RNN wants to retrieve information from the DB, it attends to the relevant rows and columns of the DB by generating attention key embeddings. While matching (to get the attention scores), in the case of non-NE information in the DB, their neural embeddings are matched with the key embeddings directly. For the NE information in the DB, exact matches are done using the NE values retrieved from the *NE-Table*, which in turn are retrieved by matching the attention key embeddings with the key embeddings of the NEs in the *NE-Table*.

the different attributes associated with it. For example, in the restaurant reservation domain, each row of the table corresponds to a new restaurant and the columns correspond to restaurant attributes such as cuisine, location, phone, address etc. For the course selection advising domain, each row corresponds to a course and the columns correspond to course attributes, such as course number, course name, instructor name etc. Each column of the table has a column heading, which labels the attribute of that column. These headings are part of the vocabulary and have neural embeddings that are learned during the dialog and retrieval process. The information within the DB is represented in 2 ways. If it is in a column which has been identified as containing NE values, then it is represented by its exact value (not a part of the vocabulary). The information in the non-NE columns become a part of the vocabulary are represented by their neural embeddings that comes from the vocabulary encoding matrix.

In order to retrieve a particular cell from the table, the system needs to find the correct column and row corresponding to it. The DB retrieval module does that by generating 3 different attention key embeddings (vectors): Attention over Columns for Columns (*ACC*), Attention over Columns for Rows (*ACR*), Attention over Rows for Rows (*ARR*). Figure 2 shows the schematic of the entire retrieval process.

The column(s) that the final retrieved cell(s) belong to, are selected by matching *ACC* key embeddings with the neural embeddings of the column headings (Course Number, Instructor, Credits etc). A separate *ACC* key embedding is generated for every column heading and matched with its embeddings to provide attention scores for all the columns. For the example, *Who teaches EECS545?*, the system would want to retrieve the name of the *Instructor*. Therefore, the *Instructor* column heading alone will have high attention score and be selected. In our experiments, the attention scores are got through dot products followed by a sigmoids, which allows for multiple selections.

Now that the column(s) are chosen, the system has to select row(s), so that it can get the cell(s) it is looking for. Each row in the table contains the values (EECS545, Machine Learning, Scott Mathew etc) of several attributes (Course Number, Course Name, Instructor etc). But we want to assign attention scores to the rows based on particular attributes that are of interest to the present scenario (*Course Number* in this example). The column/attribute headings that the system has to attend to for selecting these relevant attributes are obtained by matching *ACR* (Attention over Columns for Rows) key embeddings with the neural embeddings of the different column headings.

The last step in the database retrieval process is to select the relevant rows using the *ARR* (Attention over Rows for Rows) key embedding. *ARR* is split into two parts *ARR NE* and *ARR non-NE*. In a general scenario, *ACR* can select multiple columns to represent the rows. For each selected column that is a NE column, a separate NE value is retrieved from the *NE-Table* using a separate *ARR NE* embedding for each of them. These NE values are used to do exact match search along the corresponding columns (in the NE row representations) to select the matching rows. For the non-NE columns that are selected by *ACR*, their neural embeddings are combined together along each row to get a fixed vector representation for each row in the DB (e.g. weighted sum of their embeddings, weighted by the corresponding column attention scores). This is shown in the non-NE row representation in the Figure 2. *ARR non-NE* is then used to match these representations for selecting rows. The intersection of the rows selected in the NE row representations and the non-NE row representations is the final set of selected rows.

In short, the dialog system can use neural embedding matching for non-NEs, exact value matching for NEs and therefore a combination of both to decide which rows to attend to. Depending on the number of columns and rows we match with, we select zero, one or more output cells.

For our running example, *ARR NE* is used to match with the keys in the *NE-Table* to select the row corresponding to *EECS 545* and the value *EECS 545* is returned to do an exact match over the $NE_1$ row representations (represented by the course number values). This gives us the row corresponding to *EECS 545* and hence the cell *Scott Mathew*. We could use our *NE-Table* idea with potentially many types of neural retrieval mechanisms to retrieve information from the DB. The multiple-attention based retrieval mechanism, described above, is only one such possible mechanism.

## 3.2 Structured Question Answering from DB

The task here is to retrieve answers (single cell in a table) from a DB in response to structured one line questions. We used the details of course offerings at a University to create these question-answer pairs. Each row in the DB table corresponds to a unique course, and the different columns correspond to the different attributes of the course.

Example structured question-answer pair:
**Q**: Course Number EECS545 Credits?   **A**: 4

This is a simple task where 400 question-answer pairs were used for training and 100 question-answer pairs were used for testing. The DB is a single table of 100 rows and 4 columns (Course Number, Course Name, Department, Credits). The experiments were performed with two models. The first model uses the proposed NE idea to handle the NEs (*With-NE-Table*), which are course numbers, and course names in this task. The other (*W/O-NE-Table*) model does not distinguish NEs from normal words. In the *W/O-NE-Table* model, all words including the NEs that occur in the questions and those that occur in the DB are part of the vocabulary and have individual word representations. Both models use a simple RNN to encode the question and the multiple-attention based retrieval mechanism discussed above to retrieve answers. During training the different retrieval attentions were trained by giving the ground truth labels.

For the example question above, both the models would have to attend to the column *Credits* using *ACC* (Attention over Columns for Columns) key. For selecting the attributes essential for row matching, the models would have to attend to the column *Course Number* using *ACR* (Attention over Column for Rows) key. In the Model *W/O-NE-Table*, each row will be represented by the neural representation associated with the different course numbers, as all the different course numbers are part of the vocabulary. The *ARR non-NE* key will then be used for matching and return the selected row. For *With-NE-Table* model, the *Course Number* column will be identified as a NE column and hence the *ARR NE* key matching will happen in the *NE-Table*. The returned NE value, *EECS545*, will be used to do an exact match with the NE row representations. Note that in the model *With-NE-Table*, the rows of the DB table are going to be represented with exact course number values (no neural representations).

The test accuracy for the model *W/O-NE-Table* is 81% and 100% for the model *With-NE-Table*. The train accuracy for both the models was 100%. We found that the main difference between the train and test set for this task, is the presence of new NEs in the test set. This suggests that the 19% drop

in performance mainly comes from the NEs encountered in the questions during test time which were not seen during training time. For the model *W/O-NE-Table*, these are the NEs that are in the DB, and hence are part of the vocabulary, but have random representations which did not change during the training time. The task was specifically constructed to be simple and with a small table to show that, even in this very simple task where the *W/O-NE-Table* model achieves 100% accuracy at training time, its test accuracy is affected significantly due to new NEs that come in the questions during the test time. However, this does not pose a problem for our model *With-NE-Table*. The *With-NE-Table* model can also easily scale to large datasets with thousands of NEs without any drop in performance.

## 3.3 GOAL-ORIENTED DIALOG TASKS

We test our idea in extended versions of three dialog bAbI tasks [1] from Bordes & Weston (2016b) to evaluate our idea's usefulness in different aspects of goal-oriented dialog. Dialog bAbI dataset has 5 goal-oriented dialog tasks - Task 1: Issuing API calls, Task 2: Updating API calls, Task3: Displaying Options, Task 4: Providing extra information and Task5: Conducting full dialogs (which is a combination of tasks 1-4). Each of the four tasks (1-4) test different capabilities required in a general (commonly required) goal-oriented dialog system. The system is evaluated in a retrieval setting. At each turn of the dialog, the system has to select a candidate response from a list of possible candidates. Issuing an *api_call* corresponds to selecting a response which starts with *api_call* followed by the argument values. We evaluate our idea on extended versions of task 1, 2 and 4 (with the extensions described below). Task 3 requires the agent to learn to sort the retrieved restaurants based on their ratings and then display the options. As shown in Bordes & Weston (2016b), it is tough for the agent to learn this sorting with the limited amount of data. Their model achieves close to 0% accuracy on full dialog for this task. So we decided to skip task 3 and hence task 5 (which includes task 3 dialogs), as we are more interested in testing our idea to handle NEs, rather than trying to solve the sorting problem.

### 3.3.1 EXTENDED DIALOG BABI TASK 1: ISSUING API CALLS AND RETRIEVING INFORMATION

In the original bAbI task 1, the conversation between the system and the user involves getting information necessary to make an *api_call* and an issue of *api_call* with the appropriate argument values. We extend this task as follows. In the extended task, the system has to perform an additional task. Once the system determines that the next utterance is an *api_call*, the system also has to perform the actual query on the database to retrieve the restaurant details which match user preferences. In the original task, the system simply responds with an utterance *api_call* along with user preferences and the database retrieval is not considered. In the extended task the system is evaluated on three things: having conversation with the user, issuing *api_call* and retrieving the correct information from the DB. The DB is represented as a single table, with each row corresponding to a unique restaurant. The different columns of the table correspond to different information about the restaurant, such as its cuisine, location, address etc.

Example:

**Original dialog bAbI task 1:**
U: Hi !
```
S: Hello, what can I help you with
today ?
```
U: I'd like to book a table for six people in an expensive price range with British food.
```
S: I'm on it
S: Where should it be ?
```
U: In London
```
S: Ok, let me look into some options
for you.
S: api_call British, London, Six,
Expensive
```

**Extended dialog bAbI task 1:**
U: Hi !
```
S: Hello, what can I help you with
today ?
```
U: I'd like to book a table for six people in an expensive price range with British food.
```
S: I'm on it
S: Where should it be ?
```
U: In London
```
S: Ok, let me look into some options
for you.
S: api_call
S (Retrieved Information):
{The_Place, british, london, ..., }
{The_Fancy_Pub, british, london, ..., }
```

---

[1] The extended versions of the dialog bAbI tasks 1,2 and 4 will be made publicly available online.

### 3.3.2 EXTENDED DIALOG BABI TASK 2: UPDATING API CALLS AND RETRIEVING INFORMATION

In the original bAbI task 2, just like bAbI task 1, the conversation between the system and the user involves getting the information necessary to make an *api_call* followed by an issue of *api_call*. In task 2 however, after this, the user might ask the system to update his/her request for information by changing some of their preferences. The system has to take this into account and make an updated *api_call* at the end of the conversation. The extension task 2 is very similar to task 1: whenever the system decides to make an *api_call*, it now has to actually do the work of the *api_call* and retrieve the information from the DB. Appendix A gives an example of the original and the extended dialog bAbI task 2.

### 3.3.3 EXTENDED DIALOG BABI TASK 4: PROVIDING INFORMATION

The original dialog bAbI task 4 starts as if the user has agreed to book a table in a particular restaurant. The system is given all the information about *only* that particular restaurant. The user can ask for the phone number, address or both. The system must learn to use the given information about the restaurant of interest to answer these questions. We make few changes in the extended version. First we do not consider information just about the particular restaurant of interest to the user, instead the system needs to search and provide an answer from the full DB of all the restaurants. The candidate responses are also changed. All the NEs in the candidate responses are now replaced with their NE type tags. For example, *The_Fancy_Pub_phone* and *Suvai_phone* are now both replaced with *NE_phone*. The system has to select these candidates with NE-type tags and later replace the tags with the actual NE values by retrieving them from the DB.

Example:

**Original dialog bAbI task 4:**
DB Information given :
NAME:The_Place, CUISINE:british,
LOCATION:london, PRICE RANGE:Expensive,
RATING:7, PHONE:The_Place_phone,
ADDRESS:The_Place_address, AVAILABILITY:six
U: Hello
```
S: Hello, what can I help you with
today ?
```
U: I'd like to book a table at The_Place
```
S: Great, let me do the
reservation.
```
U: May I have the address of the restaurant.
```
S: Here it is, The_Place_address.
```

**Extended dialog bAbI task 4:**
DB Information given : Entire DB
U: Hello
```
S: Hello, what can I help you with
today ?
```
U: I'd like to book a table at The_Place
```
S: Great, let me do the
reservation.
```
U: May I have the address of the restaurant.
```
S: Here it is, NE_address.
S: The_Place_address (retrieved
information)
```

This experiment setting is closer to how a human would do this task. When someone asks for phone number/address of a restaurant, we don't try to memorize it or figure out how one phone number is related to another phone number, etc., rather, we search for the phone number/ address in the DB and return the information to the user as part of the response.

### 3.3.4 MODEL DESCRIPTION AND RESULTS FOR THE GOAL-ORIENTED DIALOG TASKS

The model that we use for the goal-oriented dialog task experiments is an instantiation of our idea (of using *NE-Table*) in an end-to-end memory network (Sukhbaatar et al., 2015). It is similar to the model used in Bordes & Weston (2016b) paper, except that we encode the sentences using an RNN, while, they use a bag of words representation. All the previous dialog history embeddings are stored in the memory. The new user utterance embedding is used to attend over the memory to get relevant information from the memory. This is done multiple times (3 in our experiments) and the final embedding obtained is transformed and used to select both the candidate response, and to generate the key embeddings for performing the retrieval operation from the DB.

For the *With-NE-Table* model, the NE key is generated (when a NE is encountered) and stored in the *NE-Table* during the process of encoding the dialog sentences using an RNN. Since, we have access to the DB, we use that to identify the NEs along with their types. The NE-type information is given

Table 1: Results for extended dialog bAbI task 1 and 2. Accuracies in % for Test and Test Out-Of-Vocabulary (given in parenthesis).

| Task | Model | ACR | ARR non-NE | ARR NE: cuisine, location | DB-Retrieval | Per-response | Per-Dialog | Per-Dialog + DB-Retrieval |
|---|---|---|---|---|---|---|---|---|
| Task 1 | W/O-NE-Table | 100 (100) | 9.3 (2.3) | - | 9.8 (7) | 99.5 (95.7) | 97.3 (74.8) | 9.5 (4.9) |
| | With-NE-Table | 100 (100) | 98.6 (98.9) | 100,100 (100,100) | 99.0 (99.0) | 99.7 (99.8) | 98.4 (98.5) | 97.9 (98.0) |
| Task 2 | W/O-NE-Table | 100 (100) | 8.6 (7.6) | - | 0.8 (0.6) | 100 (100) | 100 (100) | 0.0 (0.1) |
| | With-NE-Table | 100 (100) | 99.3 (99.0) | 100,100 (100,100) | 99.3 (99.2) | 100 (100) | 100 (100) | 98.6 (98.4) |

to both the *NE-Table* and the *W/O-NE-Table* models. In all the experiments involving retrieval from DB, the ground truth attention labels were used for training.

**Evaluation measures' description**

- *DB-Retrieval*: Retrieval percentage accuracy for rows (task 1,2) and a particular cell (task 4).
- *Per-response*: Percentage of dialog responses that are correct.
- *Per-Dialog*: Percentage of dialogs where every dialog response is correct.
- *Per-Dialog + DB-Retrieval*: Percentage of dialogs where every dialog response and all the information retrievals involved are correct.

**Experiment details and results for extended dialog bAbI tasks 1 and 2:**
The results for task 1 and task 2 are shown in Table 1. For both the models *With-NE-Table* and *W/O-NE-Table* the DB-retrieval process happens only if the system chooses to output the *api_call* sentence.

*With-NE-Table*:
For issuing an *api_call* in tasks 1 and 2, four argument values are required - cuisine, location, price range and number of people. We consider cuisine and location to be NEs. So whenever cuisine and location names occur in the dialog, a NE key is generated on the fly and is stored in the *NE-Table* along with the NE values. Here we are interested in retrieving rows from the table. So, there is no need for ACC. We need to find the column attributes with which we want to represent the rows (*ACR*) and then use that to represent the rows. There are four column attributes that are of interest, two are NEs (cuisine and location) and two are not (price range and number of people). Once *ACR* selects these four, we move to the next step. The attention over rows is split into two parts. *ARR non-NE* attends to rows based on the non-NE attributes, which are price range and number of people. So each row in the DB is represented by its price range and number of people values (weighted vector sum) and embedding key matching is done to get the attention scores. This retrieves all the rows in the DB that match the price range and number of people values that we are looking for. The second part of the row attention is *ARR NE* where the system attends over the *NE-Table* by matching its generated key with the key present in the *NE-Table* to retrieve NE values. The selected NE values are then matched (exact-match) with the cuisine and location values in the DB to retrieve the relevant rows from the DB (the rows that have that particular cuisine and location values). The final retrieved rows are the intersection of the rows selected by both these parts.

*W/O-NE-Table*:
Here, all input words (including NEs) are part of the vocabulary. For NEs however, their embedding given to the sentence encoder is the sum of the NE word embedding and the embedding associated with its NE-type. Similar to the case above, *ACR* is used to attend to the four relevant columns. However, unlike the case above, each row is represented by the combined neural embedding representation of the all the four attribute values, cuisine, location, price range and number of people. *ARR non-NE* is used to retrieve the relevant rows.

From Table 1, we can see that both the models perform well in selecting the relevant columns, but the model *W/O-NE-Table* performs poorly in retrieving the rows, while *With-NE-Table* performs very

well. This results in *With-NE-Table* model achieving close to 100% accuracy in DB retrieval while *W/O-NE-Table* performs poorly. This is because, in the *With-NE-Table* model, the retrieving rows task is split into two simpler tasks. The NEs are chosen from the *NE-Table*, and then exact matching is used (which helps in handling OOV-NEs as well). The non-NEs, price range and number of people, have limited set of possible values (low, moderate or expensive for price range and 2,4,6 or 8 for number of people respectively). This allows the system to learn good neural representation for them and hence have high accuracy in *ARR non-NE*. Whereas in *W/O-NE-Table* model, *ARR non-NE* involves the neural representations of cuisine and location values as well, where a particular location and cuisine value will occur only a few number of times in the dataset. In addition to that, new cuisine and location values can occur during the test time (Test OOV dataset, performance shown in parenthesis).

For the dialog part (which does not involve the DB retrieval aspect) of extended tasks 1 and 2, the system utterances do not have any NEs in them. However, the user utterances contain NEs (cuisine and location that the user is interested in) and so the system has to understand them in order to select the right system utterance. The accuracy in performing the dialog (by selecting responses from candidate set) is similar for both the models on the normal test set. However, in the OOV-test set, for task 1, where the system has to maintain the dialog state to track which attribute values have not been provided by the user yet, *W/O-NE-Table* model seems to get affected, while the *With-NE-Table* model is robust to that. While *W/O-NE-Table* gets a Per-Dialog accuracy of 74.8% in the OOV-test set, *With-NE-Table* is able to get 98.5%.

**Experiment details and results for extended dialog bAbI task 4:**
The results for task 4 are shown in table 2. Here, DB-retrieval happens only when the system's output sentence has a general tag/placeholder in it, which needs to be replaced by some information from the DB. As mentioned in the task description, all the NE values are replaced with their respective NE-type tags in the candidate responses. This greatly reduces the number of possible candidate responses in the dataset ('Here it is The_place_phone' and 'Here it is Suvai_phone' will now be both 'Here it is NE_phone'). So, in order to give some information such as the phone number, the system has to select a general candidate response that gives phone number (e.g. Here it is NE_phone) and then select the exact NE value to put in separately from the DB (e.g., The_place_phone). This setting is similar to the *system action templates* proposed in Hybrid Code Networks from (Williams et al., 2017). In this task, the restaurant name, phone number and address are considered as NEs. In a general generation setting, the non-NE part of the response is generated word by word by the decoder and the NE can be retrieved and inserted into the output sentence

*With-NE-Table*:
In task 4, the user tells the system the restaurant in which he/she wants to book a table. The restaurant name, which is a NE, is stored in the *NE-Table* along with it's generated key. When the user asks for information about the restaurant such as, phone number, the NE restaurant name stored in the NE table is selected and used for retrieving its corresponding phone number from the DB. For this particular case, *ACC* attends over the column *Phone* and *ACR* attends over *Restaurant Name*. Since the column selected by *ACR* is a NE column, the NE value (here the actual restaurant name given by the user) is retrieved using *ARR NE* from the *NE-Table*. The retrieved NE value is used to do an exact match over the DB column selected by *ACR* to select the rows. The cell that intersects the selected row and the column selected by *ACC* is returned as the retrieved information and used to replace the NE type tag in the output response.

*W/O-NE-Table*:
Here, all input words (including NEs) are part of the vocabulary and for NEs, their embedding given to the sentence encoder is the sum of the NE word embedding and the embedding associated with its NE-type. The candidate response retrieval (dialog) is same as the above model and the column attentions are also similar. However, the models differ with respect to attention over rows. Since NEs are not treated special here, attention over rows happens through *ARR non-NE*. For this task, when *ACR* is selected correctly (restaurant name), each row will be represented by the neural embedding representation of its restaurant names. *ARR non-NE* generates a key to match these neural embeddings to attend to the row corresponding to the restaurant name mentioned by the user.

From table 2, we observe that the dialog performance (retrieving candidate responses) for both models is very good and similar, but the *W/O-NE-Table* model fails in row retrieval while the *With-NE-Table* performs well. The difficulty for the *W/O-NE-Table* model to retrieve rows comes from

Table 2: Results for extended dialog bAbI task 4. Accuracies in % for Test and Test Out-Of-Vocabulary (given in parenthesis).

| Model | ACR | ACC | ARR non-NE | ARR NE | DB-Retrieval | Per-response | Per-Dialog | Per-Dialog + DB-Retrieval |
|---|---|---|---|---|---|---|---|---|
| W/O-NE-Table | 100 (100) | 100 (100) | 0.0 (0.0) | - | 0.0 (0.0) | 100 (100) | 100 (100) | 0.0 (0.0) |
| With-NE-Table | 100 (100) | 100 (100) | - | 100 (100) | 100 (100) | 100 (100) | 100 (100) | 100 (100) |

Table 3: Performance comparison of our model in the extended dialog bAbI tasks, with a baseline model in the original bAbI tasks. Accuracies in % for Test and Test Out-Of-Vocabulary (given in parenthesis).

| Task/Dataset | Model | Evaluation-Measure | Task 1 | Task 2 | Task 4 |
|---|---|---|---|---|---|
| Original bAbI tasks | Baseline (MemN2N + match type + RNN encoding) | Per-Dialog | 100 (100) | 99.9 (50.6) | 100 (100) |
| Extended bAbI tasks | With-NE-Table | Per-Dialog + DB-Retrieval | 97.9 (98.0) | 98.6 (98.4) | 100 (100) |

the need to learn neural representations for all restaurant names, that can be used later to match and retrieve the same.

**Performance comparison with the original dialog bAbI tasks:**
The extended dialog bAbI tasks that we present results for above require the dialog system to do strictly more work compared to the original dialog bAbI tasks. For tasks 1 and 2, while in the original version of the task, the dialog system has to have a conversation with the user and issue *api_call*s along with the arguments, in the extended version, the dialog system also has to retrieve the relevant information from the DB (in addition to having the conversation and issuing *api_call*s). For task 4, in the original task the dialog system is given just the details of the particular restaurant that the user will ask for. The dialog system has to find the correct attribute from it, depending on what the user asks for. In the extended version of task 4, the dialog system is not given information just about the restaurants the user is interested in, instead it has to use the restaurant name that the user provides during the conversation to retrieve the relevant information that the user asks for, from the full DB and use it in dialog.

Table 3 compares the performance of the *With-NE-Table* model in the extended bAbI tasks with that of a baseline method on the original bAbI tasks. The baseline method here is an end-to-end memory network with RNN encoding for its sentences (similar to the architecture used for *With-NE-Table* model), without any NE-Table or retrieval mechanism. The baseline method is also given match type features which gives extra information about the NE type when a NE occurs (as done in Bordes & Weston (2016b)). The performance that we report here for the baseline model is higher than that reported in Bordes & Weston (2016b). This is probably because we use RNN encoding for sentences, while they use bag of words representation for sentences.

We compare our method's accuracy (in the extended tasks) in getting every dialog response and every information retrieval (per-Dialog + DB-Retrieval) in a dialog correct with the baseline model's accuracy (in the original task) in getting every dialog response (Per-Dialog) in a dialog correct. Though not a strictly fair comparison for our model, from Table 3, we observe that the performance of our *With-NE-Table* model in the extended bAbI tasks is as good as the performance of the baseline model in the original bAbI tasks. In the case of bAbI task 2 OOV test set, the performance of the *With-NE-Table* model is actually much higher compared to the baseline model (98.4% vs 50.6%).

## 4 RELATED WORKS

There are several papers in the Question Answering (QA) domain that focus on DB retrieval with neural methods and suffer from the Named Entity (NE) issues described in the Introduction. Some

end-to-end systems (Neelakantan et al., 2015; Yin et al., 2015) were proposed to transform a natural language question/query to a program that could run on DBs, but those approaches are only verified on small or synthetic databases. Other papers dealing with large knowledge bases (KB) usually rely on entity linking techniques (Cucerzan, 2007; Guo et al., 2013), which links entity mentions in texts to knowledge base queries. For example, in knowledge base QA papers (Yih et al., 2015; Yin et al., 2016; Yu et al., 2017), the text spans in questions are compared with KB entity names at the character-level for entity linking; then after the linked entities have their properties extracted, the corresponding text spans are replaced with special NE tags for further text processing like KB relation extraction. Recently, Liang et al. (2016) extended end-to-end neural methods to question answering over knowledge base, which could handle large KB and large number of entities. However, their method still relies on entity linking (Yang & Chang, 2015) to generate a short list of entities linked from text spans in questions in advance. Yin et al. (2015) propose 'Neural Enquirer', a neural network architecture similar to the neural retrieval mechanism used in this work, to execute natural language queries on DB. While using the Neural Enquirer, they keep the randomly initialized embeddings of the NEs fixed as a way to handle NEs and OOV words. We could potentially use our idea of *NE-Table* with a Neural Enquirer to retrieve information from tables.

There has been a lot of recent interest in end-to-end training of dialog systems (Vinyals & Le (2015); Serban et al. (2016); Lowe et al. (2015); Kadlec et al. (2015); Shang et al. (2015)). Research on this topic tends to focus on large-scale training corpora such as movie subtitles, social media chats, or technical support logs. For large corpora it is natural to use supervised training techniques where the Recurrent Neural Networks (RNNs) attempt to replicate the recorded human utterances. However, there are also approaches that envision training via reinforcement learning techniques, given a suitably defined reward function in the dialog (Wen et al. (2016); Su et al. (2015b;a)). In more recent work on end-to-end learning of task-oriented dialog such as Bordes & Weston (2016a); Dodge et al. (2016) this paradigm is extended to decompose the main task into smaller tasks each of which must be learned by the agent and composed to accomplish the main task. Williams & Zweig (2016) use an LSTM model that learns to interact with APIs on behalf of the user. Weston (2016) (bAbI-dialog) combines dialog and reasoning to explore how an agent can learn dialog when interacting with a teacher. Guo et al. (2017) apply reinforcement learning and supervised learning in interactive reasoning tasks. Dhingra et al. (2017) use reinforcement learning to build the KB look-up in task-oriented dialog systems. But the look-up actions are defined over each entity in the KB and is therefore hard to scale up. Most of these papers actually do not discuss the issue of interacting with NEs though they are present.

In our work, we showed using simple tasks the usefulness in identifying and handling them (NEs) in the way proposed. Recently, a new hybrid approach was also proposed by (Williams et al., 2017). They proposed Hybrid Code Networks which combine an RNN with domain-specific knowledge encoded as software and system action templates. They achieved state-of-the-art performance on the Facebook bAbI dataset, but their approach involves a developer writing domain-specific software components e.g. the dialog state is maintained in the code. For certain tasks such as Machine Translation and summarization, neural copying mechanisms (Gulcehre et al. (2016), Gu et al. (2016)) have been proposed as means to handle OOV words and NEs. Here, the system has the option to copy certain words from the input to the output. We could use our *NE-Table* idea along with such copying mechanisms for cases like dialog generation.

## 5    CONCLUSION AND FUTURE WORK

In this paper we proposed a method for interacting with NEs in a goal-oriented dialog setting, which typically also involves interacting with a DB. Our experiments show that, handling NEs in the proposed way can indeed work and be useful for certain aspects of goal-oriented dialog. In future, we are interested in testing the *NE-Table* idea with other retrieval mechanisms such the 'Neural Enquirer' proposed in (Yin et al., 2015) which can work with multiple tables, instead of the multiple-attention mechanism used in our work. Our proposed idea is generic enough to be thought of as follows: in many NLP tasks, goal-oriented dialogs in particular, there are places where we are interested in encoding the meaning and relation of words/sentences by learning neural embedding representations for them (for understanding a user question for example), there are also places where we are not interested in the meaning, but rather just in the exact values (phone numbers, address etc). One reason could be that working with these exact values can make certain tasks easier and accurate

(compared to having a neural embedding for them) and another practical reason could be that learning good neural representation for these particular values that occur rarely (sometimes only during test time) is difficult. The *NE-Table* idea provides a way to work with exact values of entities but still remain within the neural learning scheme, which allows the overall task to be end-to-end trainable using back propagation. In future work, we are interested in exploring the use of this idea in a more general setting of working with both symbols and distributed representations, exploiting the benefits of both of them wherever appropriate and doing all this within a end-to-end trainable neural setting.

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

## APPENDIX A: EXTENDED DIALOG BABI TASK 2 EXAMPLE

**Original dialog bAbI task 2:**
U: Hi !
S: Hello, what can I help you with
today ?
U: I'd like to book a table for six people in an
expensive price range with British food in London.
S: Ok, let me look into some
options for you.
S: api_call British, London, Six,
Expensive
U: Instead, could it be Indian cuisine
S: Sure, is there anything else to
update ?
U: No.
S: Ok, let me look into some
options for you
S: api_call Indian, London, Six,
Expensive

**Extended dialog bAbI task 2:**
U: Hi !
S: Hello, what can I help you with
today ?
U: I'd like to book a table for six people in an
expensive price range with British food in London.
S: Ok, let me look into some
options for you.
S: api_call
S (Retrieved Information):
{The_Place, british, london, ...,}
The_Fancy_Pub, british, london, ..., }
U: Instead, could it be Indian cuisine
S: Sure, is there anything else to
update ?
U: No.
S: Ok, let me look into some
options for you
S: api_call
S (Retrieved Information):
{Suvai, Indian, london, ..., }
{Madras_Masala, Indian, london, ..., }

