# OpenReview forum: "A Neural Method for Goal-Oriented Dialog Systems to interact with Named Entities"
_ICLR.cc/2018/Conference — Reject_

### Official Review · AnonReviewer3 · 2017-11-27
**Recognizing and Updating NE Embeddings on the Fly**

**Rating:** 6
**Confidence:** 3

**Review:**

The paper proposes to generate embedding of named-entities on the fly during dialogue sessions. If the text is from the user, a named entity recognizer is used. If it is from the bot response, then it is known which words are named entities therefore embedding can be constructed directly. The idea has some novelty and the results on several tasks attempting to prove its effectiveness against systems that handle named entities in a static way.

One thing I hope the author could provide more clarification is the use of NER. For example, the experimental result on structured QA task (section 3.1), where it states that the performance different between models of With-NE-Table and W/O-NE-Table is positioned on the OOV NEs not present in the training subset. To my understanding, because of the presence of the NER in the With-NE-Table model, you could directly do update to the NE embeddings and query from the DB using a combination of embedding and the NE words (as the paper does), whereas the W/O-NE-Table model cannot because of lack of the NER. This seems to prove that an NER is useful for tasks where DB queries are needed, rather than that the dynamic NE-Table construction is useful. You could use an NER for W/O-NE-Table and update the NE embeddings, and it should be as good as With-NE-Table model (and fairer to compare with too).

That said, overall the paper is a nice contribution to dialogue and QA system research by pointing out a simple way of handling named entities by dynamically updating their embeddings. It would be better if the paper could point out the importance of NER for user utterances, and the fact that using the knowledge of which words are NEs in dialogue models could help in tasks where DB queries are necessary.

---

> ### Author Response · Authors · 2017-12-29
> **Please find the answers to individual questions below**
>
> R:"One thing I hope the author could provide more clarification is the use of NER. For example, the experimental result on structured QA task (section 3.1), where it states that the performance different between models of With-NE-Table and W/O-NE-Table is positioned on the OOV NEs not present in the training subset. To my understanding, because of the presence of the NER in the With-NE-Table model, you could directly do update to the NE embeddings and query from the DB using a combination of embedding and the NE words (as the paper does), whereas the W/O-NE-Table model cannot because of lack of the NER. This seems to prove that an NER is useful for tasks where DB queries are needed, rather than that the dynamic NE-Table construction is useful. You could use an NER for W/O-NE-Table and update the NE embeddings, and it should be as good as With-NE-Table model (and fairer to compare with too)."
>          It is totally true that NER is useful for these kind of tasks, and we utilise it. But just the NER alone is not enough. For example, in a dialog there could be multiple NEs of the same type occurring at different places and we need to choose the right one, say for a DB query. This can be done using the dynamic NE-table idea and it also a differentiable process and hence can be learnt using back propagation. Just identifying the NEs alone is not enough.
>         We do accept that for simple structured synthetic tasks such as the QA task of ours one could write a simple rule based system to utilise NER and do the task perfectly. Infact for the bAbI task, as shown in the original paper, one can build a rule based system (which are brittle ofcourse) to get 100% accuracy on all tasks, since the data is a synthetic simulated one, but this is not true for real data. For more sophisticated tasks, where it's not possible to solve the task by writing a set of rules, our approach for handling named entities provides a first of its kind solution and addresses problems associated with learning embeddings for rare NEs and handling OOV NEs. It also gives a way to work with exact NE values, within the neural learning framework.
>          Also, for all our dialog tasks in the bAbI dialog dataset, we do give the NE type information to the baseline W/O-NE table model as well.
>
> R:"It would be better if the paper could point out the importance of NER for user utterances, and the fact that using the knowledge of which words are NEs in dialogue models could help in tasks where DB queries are necessary."
>         We totally agree with this point and we have added this to our updated version of the paper.

---

### Official Review · AnonReviewer1 · 2017-11-27
**Poor presentation**

**Rating:** 4
**Confidence:** 3

**Review:**

The paper addresses the task of dealing with named entities in goal oriented dialog systems. Named entities, and rare words in general, are indeed troublesome since adding them to the dictionary is expensive, replacing them with coarse labels (ne_loc, unk) looses information, and so on. The proposed solution is to extend neural dialog models by introducing a named entity table, instantiated on the fly, where the keys are distributed representations of the dialog context and the values are the named entities themselves. The approach is applied to settings involving interacting to a database and a mechanism for handling the interaction is proposed. The resulting model is illustrated on a few goal-oriented dialog tasks.


I found the paper difficult to read. The concrete mappings used to create the NE keys and attention keys are missing. Providing more structure to the text would also be useful vs. long, wordy paragraphs. Here are some specific questions:

1. How are the keys generated? That are the functions used? Does the "knowledge of the current user utterance" include the word itself? The authors should include the exact model specification, including for the HRED model.

2. According to the description, referring to an existing named entity must be done by "generating a key to match the keys in the NE table and then retrieve the corresponding value and use it". Is there a guarantee that a same named entity, appearing later in the dialog, will be given the same key?  Or are the keys for already found entities retrieved directly, by value?

3. In the decoding phase, how does the system decide whether to query the DB?

4. How is the model trained?

In its current form, it's not clear how the proposed approach tackles the shortcomings mentioned in the introduction. Furthermore, while the highlighted contribution is the named entity table, it is always used in conjunction to the database approach. This raises the question whether the named entity table can only work in this context.

For the structured QA task, there are 400 training examples, and 100 named entities. This means that the number of training examples per named entity is very small. Is that correct? If yes, then it's not very surprising that adding the named entities to the vocabulary leads to overfitting. Have you compared with using random embeddings for the named entities?

Typos: page 2, second-to-last paragraph: firs -> first, page 7, second to last paragraph: and and -> and

---

> ### Author Response · Authors · 2017-12-28
> **Please find answers to individual questions below**
>
> Following your suggestion, we have rewritten the proposed solution in section 2 adding equations and modifying the figure to give the concrete mappings used to create the NE keys. We have tried to split the core idea from the retrieval mechanism used in the experiments in the text and have explained the retrieval mechanism separately in section 3.1
>
> R:"1. How are the keys generated? That are the functions used? Does the "knowledge of the current user utterance" include the word itself?"
>     When using an RNN sentence encoder, the exact mechanism for generating an embedding for a NE is shown in equation 1 in section 2. It is also shown in figure 1. In words, when the encoder RNN encounters a NE, the representation of the dialog so far, the sentence representation of the current user utterance so far and the NE type information are used (a linear transformation) to generate a neural embedding on the fly and is stored in the NE-Table as key along with the NE associated with it stored as the value.
> The knowledge of the current NE word is given by its NE-type.
>
> R:"2. According to the description, referring to an existing named entity must be done by "generating a key to match the keys in the NE table and then retrieve the corresponding value and use it". Is there a guarantee that a same named entity, appearing later in the dialog, will be given the same key?  Or are the keys for already found entities retrieved directly, by value?"
>     A new key is generated for each named entity that comes during a dialog, irrespective of whether they have occurred before in the dialog (hence already in the NE table) or not.
>
> R:"3. In the decoding phase, how does the system decide whether to query the DB?"
>     The system has to make a decision whether it has to query the DB or not. The exact way and place where this is done is task dependent. For example, in the QA task, query to the DB happens always. In the case of the bAbI dialog task 1 and 2, query to the DB happens only if the agent chooses to output the “api_call’’ sentence. In dialog task 4, the DB query happens whenever the output system utterance has a NE_tag as a part of it (hence requires some information from the DB).
>
> R:"4. How is the model trained?"
>     In all our experiments the model is trained in a fully supervised way, along with the labels for column and row attentions (for the DB retrieval mechanism).
>
> R:"In its current form, it's not clear how the proposed approach tackles the shortcomings mentioned in the introduction. Furthermore, while the highlighted contribution is the named entity table, it is always used in conjunction to the database approach. This raises the question whether the named entity table can only work in this context."
>     We have a method that can learn to store and point to any NE that has occurred in the dialog so far, depending upon the requirement of the downstream task. This gives neural way of interacting with NEs that does not have the following issues: Explosion in vocabulary size, issues associated with not learning good embeddings as a particular NE occurs only few times in a dataset, loss of information and inability to refer to particular NEs of the same type, which happens when NEs are replaced by their NE type tags, issues with OOV NE words. The idea is general enough to be applied in any NLP task that involves NEs. Here we focus on goal-oriented dialog, which almost always has an external DB and has the set of issues mentioned above. We are working on experiments which don't involve a DB and will share more details on future work.
>
> R:"For the structured QA task, there are 400 training examples, and 100 named entities. This means that the number of training examples per named entity is very small. Is that correct? If yes, then it's not very surprising that adding the named entities to the vocabulary leads to overfitting. Have you compared with using random embeddings for the named entities?"
>      NEs in general have this issue of not being able to learn good neural embeddings for them since the individual NEs occur only few number of times in a dataset. As you have pointed out, that is very much true here as well. Here the whole dataset dominated by lot of NEs, but the training examples per NE is similar to a real world scenario.
> The point that we were trying to emphasize was about the new NEs that come during the test time in the questions, which the system has not seen in the questions during the training time. The embeddings of these new NEs that come during the test time alone, have not been trained and hence remain random as they were initialised. In a general task, this might be confusing to a system that tries to interpret or work with their embeddings. Following your suggestion, we did an experiment (Structured QA task) with random embeddings for NEs and fixing it throughout, the test performance (accuracy) is still 82 %, compared to the 100% accuracy which we obtain using the NE-table idea.

---

### Official Review · AnonReviewer2 · 2017-11-27
**Some good ideas, but lack of detailed explanation impacts understanding**

**Rating:** 3
**Confidence:** 3

**Review:**

Properly capturing named entities for goal oriented dialog is essential, for instance location, time and cuisine for restaurant reservation. Mots successful approaches have argued for separate mechanism for NE captures, that rely on various hacks and tricks. This paper attempt to propose a comprehensive approach offers intriguing new ideas, but is too preliminary, both in the descriptions and experiments.

The proposed methods and experiments are not understandable in the current way the paper is written: there is not a single equation, pseudo-code algorithm or pointer to real code to enable the reader to get a detailed understanding of the process. All we have a besides text is a small figure (figure 1). Then we have to trust the authors that on their modified dataset, the accuracies of the proposed method is around 100% while not using this method yields 0% accuracies?

The initial description (section 2)  leaves way too many unanswered questions:
- What embeddings are used for words detected as NE? Is it the same as the generated representation?
- What is the exact mechanism of generating a representation for NE EECS545? (end of page 2)
- Is it correct that the same representation stored in the NE table is used twice? (a) To retrieve the key (a vector) given the value (a string)  as the encoder input. (b) To find the value that best matches a key at the decoder stage?
- Exact description of the column attention mechanism: some similarity between a key embedding and embeddings representing each column? Multiplicative? Additive?
- How is the system supervised? Do we need to give the name of the column the Attention-Column-Query attention should focus on? Because of this unknown, I could not understand the experiment setup and data formatting!

The list goes on...

For such a complex architecture, the authors must try to analyze separate modules as much as possible. As neither the QA and the Babi tasks use the RNN dialog manager, while not start with something that only works at the sentence level

The Q&A task could be used to describe a simpler system with only a decoder accessing the DB table. Complexity for solving the Babi tasks could be added later.

---

> ### Author Response · Authors · 2017-12-28
> **Please find answers to individual questions below**
>
> Following your suggestion, we have rewritten the section that explains the proposed method (Section 2) (along with equations to show the exact process of NE key generation) and the section that explains the retrieval mechanism used for the experiments (Section 3.1). We have also tried to modify the figures for easier and detailed understanding. We hope the modified text read in coordination with the figure should make it understandable now.
>
> R: "What embeddings are used for words detected as NE? Is it the same as the generated representation?"
>         Yes, the new generated key embedding for the NE is fed to the sentence encoder as the NE’s word embedding.
>
> R:"What is the exact mechanism of generating a representation for NE EECS545?"
>         When using an RNN sentence encoder, the exact mechanism for generating an embedding for a NE is shown in equation 1 in section 2. It is also shown in figure 1. In words, when the encoder RNN encounters a NE, the representation of the dialog so far (h^d_t−1 ), the sentence representation of the current user utterance so far (h^x,enc_t,i−1 ) and the NE type information are used (a linear transformation) to generate a neural embedding (z^ne_i ) on the fly and is stored in the NE-Table as key along with the NE x_t,i associated with it stored as the value.
>
> R: "Is it correct that the same representation stored in the NE table is used twice? (a) To retrieve the key (a vector) given the value (a string)  as the encoder input. (b) To find the value that best matches a key at the decoder stage?"
>         No. It is true that the representation stored in the NE table is used to find the value that best matches a key at the decoder stage. In the encoder side, when the encoder encounters a NE, it generates a new key (using the knowledge of the dialog so far and the NE type) for it and stores it in the NE table (even if this particular NE has come before in the dialog)
>
> R:"Exact description of the column attention mechanism: some similarity between a key embedding and embeddings representing each column? Multiplicative? Additive?"
>         We use dot product followed by sigmoid for finding the similarity and calculating the attention score.
>
> R:"How is the system supervised? Do we need to give the name of the column the Attention-Column-Query attention should focus on?"
>         Yes, as mentioned in the paper, in all our experiments, the system is supervised. We give labels to the column and row attentions.
>
> R:"For such a complex architecture, the authors must try to analyze separate modules as much as possible."
>         We have tried to incorporate this suggestion into our new modified version. In Section 2: “Proposed solution”, we focus only on our core idea of usage of NE tables, without getting into the dialog manager or the retrieval mechanism used in our experiments. Later in Section 3.1, we explain the retrieval mechanism used in our experiments.

---

### Author Response · Authors · 2017-12-28
**Along with other minor changes, the “Proposed Solution” section (2) and experimental setup section (3.1) have been rewritten for better understanding**

First, we would like to thank the reviewers for their valuable reviews. The main issue, as we understand, seems to be with the writing and hence, the understandability of the proposed idea. We have made our sincere efforts in rewriting the paper, mainly the “Proposed Solution” section (2) and experimental setup section (3.1) to make things more clear and explicit. Though we answer the specific questions raised in the reviews below, we do request the reviewers to go through the paper once more for a more clear explanation of the idea and experiments.

---

### Decision · Program_Chairs · 2018-01-29
**ICLR 2018 Conference Acceptance Decision**

**Decision:**

Reject

**Comment:**

This work deals with the important task of capturing named entities in a goal-directed setting. The description of the work and the experiments are not ready for publication; for example, it is unclear whether the proposed method would have an advantage over existing methods such as the match type features that are only mentioned in Table 3 for establishing the baseline on the original bAbI dialogue dataset, but not even discussed in the paper.